# Psychometric Testing of the Slovene Version of the Perceived Inventory of Technological Competency as Caring in Nursing

**DOI:** 10.3390/healthcare12050561

**Published:** 2024-02-28

**Authors:** Cvetka Krel, Dominika Vrbnjak, Gregor Štiglic, Sebastjan Bevc

**Affiliations:** 1Department of Nephrology, University Clinical Centre Maribor, 2000 Maribor, Slovenia; sebastjan.bevc@ukc-mb.si; 2Faculty of Health Sciences, University of Maribor, 2000 Maribor, Slovenia; dominika.vrbnjak@um.si (D.V.); gregor.stiglic@um.si (G.Š.); 3Faculty of Medicine, University of Maribor, 2000 Maribor, Slovenia

**Keywords:** technology, caring, reliability, validity, psychometric properties

## Abstract

The Perceived Inventory of Technological Competency as Caring in Nursing (PITCCN) questionnaire has been designed to measure technological competency as caring in nursing practice. It incorporates the use of technology with the fundamental principles of caring that are central to nursing. As there were no psychometrically sound instruments to quantify the concept of technological competency as caring in the Slovene language, we adapted the English version of the questionnaire to the local environment. The goal was to assess the level of psychometric properties of the PITCCN investigated in Slovene hospitals. Methods: Content validity was conducted with eight experts and quantified by the content validity index (CVI) and the modified Cohen’s kappa index. Face validity was assessed through discussions with participants from the target culture in the pilot study. To assess construct validity and internal consistency, a cross-sectional research methodology was used on a convenience sample of 121 nursing personnel from four hospitals. Principal component analysis (PCA) was used to examine construct validity, while Cronbach’s alpha and adjusted item-total correlations were used to measure internal consistency. Results: The content and face validity of PITCCN were adequate. The scale validity index (S-CVI) was 0.97. Cronbach’s α was 0.92, and subscale reliabilities ranged from 0.810 to 0.925. PCA showed four components, which explained more than 73.49% of the variance. Conclusions: The Slovenian version of PITCCN (PITCCN_SI) has good psychometric properties.

## 1. Introduction

The development of technology in healthcare, such as electronic health systems for collecting and storing healthcare information, has become more prevalent and beneficial to healthcare organizations. Electronic health systems allow it to keep patient records and other health information in a way that can be shared between patients, healthcare workers, and hospitals [1]. The integration of technology into nursing practice involves the utilization and analysis of extensive data. Nurses create voluminous patient medical information in electronic health records when they document patients’ care, such as patient’s responses to treatments, symptoms, pain, etc. [2]. Accurate, up-to-date information and reliable patient data allow effective decision-making and appropriate action during nursing care provision [3,4,5] and make nursing care visible [6]. Technology is the tool that should help nurses in the first place to provide safe, quality, and patient-centered nursing care [7]. However, the use of technology also raises the potential concern that the focus might shift from caring to using technology itself [8]. Nurses need training, support, and funding to conduct research that uses data [9].

Nursing theories rooted in the concept of caring serve as guiding frameworks for nursing practice by aiding nurses in clarifying their beliefs and beliefs about human health processes, thereby shaping their approaches to patient care [10]. Integrating nursing theory into practice contributes to the enhancement of nursing care quality [11]. Locsin [12] underlined the importance of theory-based nursing practice, asserting that using technology and caring in practice leads to harmonious nursing assistance through technology. The “Technological Competence as Care in Nursing” (TCCN) theory is one of the nursing theories related to caring and using technologies in nursing. Locsin [8] clarified why nurses need to be technologically competent. Technological competency is a term referring to caring in nursing where technology and caring coexist. It is used to understand persons as caring and more fully as participants in their care rather than as objects of care [12]. He also incorporated the theorist Carper’s four modes of knowing (ethical, aesthetic, empirical, and personal) [13], adding that the use of technologies in nursing necessitates technological knowing [8]. Technological knowing is another form of knowing in nursing; it entails the competent use of care technology to get to know patients as a whole, which is unique (with their hopes, dreams, and desires), whole (body, mind, and soul), and perfect (regardless of sickness) [14]. With a focus on operationalizing these concepts, Locsin [15] developed the Technological Caring Instrument (TCI) to measure technological caring. Parcells and Locsin [16] revised the TCI. They developed the Technological Competency as Caring in Nursing Instrument (TCCNI), which incorporates knowing persons more fully in conjunction with caring with/through technology and measures nurses’ ways of thinking about caring and technology [17,18,19]. The questionnaire Perceived Inventory of Technological Competency as Caring and Nursing (PITCCN) was developed by Kato et al. [20], revised by Miyamoto et al. [21], and validated by Ito et al. [22]. The PITCCN consists of four factors, which are measured with this scale: (1) training of nurses to provide optimal care, (2) intentional and ethical nursing of person, (3) utilization of information obtained from technology and continuous knowing, and (4) empirical knowledge and knowing wholeness of persons.

The surveys of nurses were carried out in intensive care units in Japan to detect perceptions and practice situations of TCCN theory and technological competency. The PITCCN was targeted and tested on its contribution to practice and can measure technological competency as an expression of caring in nursing [22,23]. Therefore, we also used this questionnaire. The key idea of PITCCN was that the empirical, personal, ethical, and aesthetic methods of knowing that are essential to understanding patients as a whole increase the likelihood of getting to know patients [21]. In PITCCN, empirical knowing was defined as a method of understanding patients through the use of technologies that contribute to scientific knowledge to support interventions in nursing care [22]. In our survey, we used the PITCCN questionnaire [22] to measure its capacity for technological competency as an expression of caring in nursing [20,22], addressing the lack of psychometrically valid and reliable instruments for evaluating technological competency as caring in the Slovene language.

To bring theory into practice, we integrated the TCCN theory and the PITCCN questionnaire into our investigation, centering on developing and implementing an electronic nursing record system (ENRS). Based on Locsin’s TCCN theory, ENRS can use a language of caring in nursing practice, surpassing the limitations of exclusively documenting physical requirements and medical conditions. ENRS can provide information about patients as a whole, including documenting the work performed by multiple healthcare providers and enabling better-coordinated care [24].

The study aims to investigate the psychometric properties of a Slovenian version of PITCCN (PITCCN_SI) by evaluating its content and construct validity through Principal Component Analysis (PCA) and internal consistency.

## 2. Materials and Methods

### 2.1. Validation concerning the Experts’ Agreement

#### 2.1.1. The Process of Translating the Questionnaire

One of the original PITCCN writers permitted us to use the instrument. The instrument by Ito et al. [22], PITCCN with 19 items, was translated from English to Slovenian, followed by back translation to ensure semantic equivalence [25] and questionnaire quality [26].

The instruments were separately translated into Slovene by two translators. One of the translators was acquainted with the research on technological competency in caring in nursing. Both then discussed the translated instrument with another researcher who is fluent in English and holds expertise in the research field. A professional translator backtranslated the instrument when preparing a consensus version. The research team and translators compared responses to the translated and original versions of the questionnaire. They eliminated any discrepancies that might indicate problems with semantic or cultural equivalence.

#### 2.1.2. Content Validity

Eight experts with a PhD or MSc degree and over three years of nursing experience assessed content validity. Their educational background encompassed nursing theories, including caring theories, and expertise in utilizing nursing technologies, such as electronic nursing record systems.

Purposive sampling was used. Experts examined the instruments for their relevance using a four-point relevance scale (4 = highly relevant, 3 = quite relevant, 2 = somewhat relevant, 1 = not relevant). A content validity index (CVI) was calculated for each item (I-CVI) and scale (S-CVI) [26,27]. Content validity indices were evaluated (I-CVI and S-CVI/Average) as good, with values of 0.78 and 0.90, respectively [28]. The I-CVI was calculated as the number of experts presenting a rating of 3 or 4 divided by the number of experts. The S-CVI was calculated as an average of the I-CVIs for all items on the scale (S-CVI/Ave). A modified kappa statistic (κ*) was computed to account for chance agreement on relevancy and use among experts [19]. The probability of chance agreement was first calculated using the formula $Pc=(\frac{N!}{A!}\times \left(N-A\right))\times 0.5^{N}$, where N is the number of experts, while A is the number of agreeing on good relevance, with ratings of 3 and 4. The following formula was then used to determine the κ* = [I-CVI − Pc]/[1 − Pc]. The value for each κ* is excellent (more than 0.74), good (between 0.60 and 0.74), or fair (between 0.4 and 0.59) [27,28].

#### 2.1.3. Face Validity and the Cultural Equivalence of Items

The face validity and cultural equivalence of items were evaluated in a pilot study involving a convenience sample of 12 experienced members of the nursing teams (registered nurses and nursing assistants). They have had more than three years of experience in nursing, have worked in one of the internal wards of Slovenian hospitals, and were proficient in using the ENRS. Participants were invited to complete the questionnaire, estimate the time needed for completion, and assess the comprehensibility (face validity) by proposing enhanced phrasing for any unclear items. Following their comments or suggested revisions, the authors engaged in discussions to determine whether the items maintained cultural significance, aiming to reach a consensus before finalizing and confirming the questionnaire.

Detailed documentation was maintained throughout the translation and adaptation process. It included records of translation, expert reviews, the pilot feedback, and any modifications made to enhance semantic and cultural equivalence. It was reviewed by a panel of experts (PhD-trained professors; one expert in research and methodological studies, two in nursing care, and one in nursing technology). The panel of experts was in contact with the original developers of the PITCCN questionnaire during this part of the process.

### 2.2. Participants, Setting, and Procedure

A cross-sectional survey was conducted to evaluate the instrument’s psychometric properties. Psychometric testing involved a convenience sample of 121 participants (registered nurses and nursing assistants) in two internal wards and two non-acute care settings from four Slovenian hospitals using the same nursing record systems. We have included nurses and nursing assistants in the study because, in Slovenia, both professional groups are members of nursing teams who actively engage in various nursing interventions and use ENRS within hospital settings. Nursing assistants undergo secondary-level education specific to their role, while nurses receive a 3-year first-cycle Bologna higher education. According to the National Institute of Public Health of the Republic of Slovenia (2020), currently, 72% (*n* = 12.959) are nursing assistants and 28% (*n* = 9.043) are nurses in Slovenia.

We included participants working in internal wards and non-acute care within hospitals who were willing to participate and utilize the identical ENRS. Gender, age, education, professional, and experience with ENRS were among the demographic features recorded by respondents. Nurses using ENRS for less than three months were excluded from the survey.

Data were collected from August 2021 to May 2022. Head nurses of two internal wards and two non-acute care settings from hospitals distributed the questionnaires prepared in sealed envelopes to the participants. They informed them of the study’s aims before administration. A total of 164 questionnaires were sent and were to be returned within four weeks. Returning completed questionnaires was regarded as consent for participation. The completed questionnaires were returned in a closed envelope to the head nurses or the researcher, who collected them and securely stored them. All questionnaires had security codes attributed to computerized records.

### 2.3. Data Analysis

The questionnaires were coded with a number (from 01 to 121) and data were exported to IBM SPSS Statistics (version 24.0 for Windows, Armonk, NY, USA) and analyzed. The question asking for numerical data (age) was not coded because participants entered the exact number into a blank box. For nominal data (gender, professional experience, education, time of use ENRS), codes were assigned arbitrarily (e.g., Female ‘2’, Male ‘1’). The responses in the PITCCN_SI were coded as ‘1’ (strongly disagree) to ‘5’ (strongly agree). Descriptive statistics were performed to describe the population’s characteristics and the PITCCN_SI results. Cronbach’s alpha was calculated for internal consistency and used as acceptable if the coefficient was above 0.7 [28]. The item-total correlation of each item and the item’s contribution were analyzed. If the correlation of an individual item with the rest of the scale was above 0.3, the item was eliminated [28,29]. PCA was used to evaluate the construct validity, explaining as much variance as possible in the data and reducing variables called components, which are linear combinations of the original variables [30]. Bartlett’s sphericity test and Kaiser–Meyer–Olkin (KMO) were used to determine how suited the data were for factor analysis. A criterion of KMO values between 0.8 and 1 indicates the sampling is adequate and that the higher the KMO value, the more appropriate the data are for factor analysis [31], and Bartlett’s test of sphericity values are significant (i.e., the significant value must be 0.05 or less) [32].

## 3. Results

### 3.1. Validation concerning the Experts’ Agreement

#### 3.1.1. The Process of Translating the Questionnaire

We encountered no issues with the translated items during the expert review, and there was no need for any modifications or deletions of items.

#### 3.1.2. Content Validity of the PITCCN_SI

Table 1 contains 19 questions, classified into four subscales from the PITCCN questionnaire by Ito et al. [22]. As described in Table 1, only one item (I continue to consider better care by reflecting on their process of care) had an I-CVI score (0.62) lower than 0.78. The S-CVI/Ave was 0.974, showing satisfactory content validity.

#### 3.1.3. Face Validity and the Cultural Equivalence

In the pilot study, the research team evaluated the cultural and semantic equivalence and comprehensibility of the items by consulting with experienced nursing team members from the target culture, including registered nurses and nursing assistants.

This process aimed to ensure that the translation of the instrument accurately preserved the meaning of each item across the cultures of interest.

The findings indicated that the items were culturally appropriate, easy to understand, and free of cultural biases. There were no semantic issues. The entire questionnaire took a mean time of 10 min to complete.

### 3.2. Sample

In total, 121 out of 164 fully completed surveys were returned (return rate = 73.78%). It was mostly women (85.1%) and nursing assistants (48.8%) who participated in the research. The mean age was 36.06 (SD = 10.85). Most participants had up to 5 years of working experience (40.5%) and 6–11 months of use of ENRS (38.8%). Other characteristics of the sample are presented in Table 2.

### 3.3. Construct Validity of PITCCN_SI

Bartlett’s test (Approx. Chi-Square = 1656.12, df = 171, *p* ≤ 0.001) and the KMO (0.89) showed acceptable values, and the data were found to be adequate to perform PCA. Table 3 shows that the PCA provided a four-component solution that explained 73.49% of the data variance. The first component was labeled as training of nurses to provide optimal care and contained seven items with factor loadings from 0.49 to 0.82. The second component, empirical knowledge and whole human knowing, had three items, with factor loadings from 0.74 to 0.85. The third component, utilization of information obtained from technology and continuous knowing, contained three items whose factor loadings ranged from 0.86 to 0.92. The fourth component, labeled as intentional and ethical nursing of a person, included six items whose factor loadings ranged from 0.63 to 0.85.

### 3.4. Internal Consistency Reliability of PITCCN_SI

The latest version of PITCCN_SI had a satisfactory Cronbach’s α for a scale of 0.925. In addition, the four subscales (nursing education to give optimal care, complete human knowing and empirical knowledge, integration of information received from ENRS, and committed and ethical nursing care) obtained from the PCA also indicated satisfactory Cronbach’s α. The Cronbach’s α for subscales was (0.901, 0.810, 0.897, and 0.921), as seen in Table 3.

## 4. Discussion

Our study found that PITCCN_SI has satisfactory psychometric properties, with 121 nurses responding. The content and face validity, reliability, and construct validity of PITCCN_SI were verified.

A version of the PITCCN_SI was developed that was semantically and culturally adapted to be used by Slovenian members of nursing teams. The process of back translation was used. Forward and backward translations are essential for the survey’s validity to achieve semantic equivalence [33] and a high-quality translation, which confirmed the accuracy of the translations for the Slovene version of the PITCCN questionnaire. The research team and translators discussed the translated and original versions of the questionnaire. They eliminated any discrepancies that might indicate problems with semantic or cultural equivalence. To guarantee the questionnaire’s suitability in representing the targeted construct, the expert panel, assessing content validity, evaluated the relevance of each question’s content, the appropriateness of language, cultural pertinence, and coverage across domains [26].

We carried out content validity, which is not often used but is recommended for psychometric testing and cultural adaptations [26,34]. In professional nursing practice, the differences relate to demographics, culture, and differences in the healthcare system. Also, the development and use of technology in nursing differ in each country [22,35]. The CVI indicates that the degree of agreement between the expert raters and the criteria for item acceptability should be no lower than 0.78 [28]. Only one of the items in our study (I continue to consider better care by reflecting on their process of care) in the instrument had an I-CVI score lower than 0.78. The experts suggested only a more comprehensible reformulation of the item in the Slovenian language, so it was not eliminated from the questionnaire and was considered in further psychometric testing, where it reached acceptable values when testing internal reliability. The average CVI for the scale showed satisfactory content validity. The content and face validity of PITCCN were found to be adequate. The research team assessed the items’ cultural equivalence and comprehensibility by discussing with participants from the target culture (registered nurses and nursing assistants) in the pilot study to improve clarity and cultural relevance. They confirmed the items were culturally appropriate, easily understood, and free from cultural bias during the pilot study. Comprehensive records were kept during the translation and adaptation process, documenting translation decisions, expert reviews, pilot study feedback, and modifications made to improve semantic and cultural equivalence. They provide transparency and can facilitate future research or validation efforts [26].

The data were found to be suitable and acceptable to perform a PCA by Bartlett’s test, and the KMO showed acceptable values. The PCA lessens the dimensionality of a dataset while preserving as much ‘variability’ (statistical data) as possible [36], and our research produced a four-component solution. The same four components had already been obtained in previous studies [22,23]. Cronbach’s alpha is an essential concept in the evaluation of assessments and questionnaires. Good Cronbach’s alpha values indicate a sufficient number of questions, interrelatedness of items, or heterogeneous constructs [37]. The PITCCN_SI version demonstrated a satisfactory Cronbach’s α for the overall scale. The final version of the original PITCCN had a satisfactory Cronbach’s a scale [22], which has also been observed in previous research [38]. The internal reliability of individual subscales was also acceptable because they were also all over the limit value of 0.7. They were also confirmed in the PITCCN questionnaire by Ito et al. [22], and there were values of individual subscales over the limit [22].

The original instrument had 23 items. The questionnaire was tested in Japanese hospitals in intensive care units. Since the original questionnaire also contained questions about managing unconscious patients in intensive care units, we decided to use the newer PITCCN questionnaire by Ito et al. [22], who retained 19 of the 23 items after factor analysis. Thus, we did not include the following items of the original PITCCN in our research: (1) I deliberately try to communicate with unconscious patients with the aim of resuscitation; (2) I empathize with what patients experience; (3) I appreciate knowing that the patient is most hopeful now; and (4) I am required to respect the privacy of unconscious patients. We found no other differences between the original and PITCCN_SI during the verification process.

Locsin’s (2005) TCCN theory underpinned our research as it articulates explicitly the idea of nursing practice being integrated with modern healthcare technology [39], where they also belong to electronic health records [39]. Theoretical starting points also supported research on electronic health records [40,41], but few surveys have been carried out. TCCN theory will guide our study in implementing the ENRS and documenting nurses’ caring behaviors in nursing, not just physical needs and health conditions. There are essential caring interventions, such as being hopeful, providing psychical care, carefully listening to patients, showing compassion, prayer, using therapeutic touch, etc. [42]. It is necessary to understand patients as complete and unique [8], which must also be considered when documenting the nursing process. Research confirmed that with the introduction of computers into each patient room, nurses spent more time with the patient, and caring interventions increased after implementing electronic health records [2,43]. However, there has been little reliable research on the pros and cons, including nurses’ caring behaviors, of its use. Nurses have been open to exploring the benefits of electronic documentation and building competency in integrating technology and caring [43]. A limitation of the study is its small sample size, as participants were recruited through convenience sampling. Additionally, the study was limited to only include a subset of hospitals. Additional research is needed to explore different settings and diverse populations. Because criterion-related validity and test–retest reliability were not investigated in this study, further psychometric evaluation is necessary.

## 5. Conclusions

The Slovenian version of the PITCCN questionnaire was developed and tested based on the TCCN theory but has not previously been used to conduct a study in Slovenia. A version of the PITCCN questionnaire similar to the original version was obtained, which was semantically and culturally adapted to Slovenian nursing care providers. The PITCCN questionnaire is a proper tool that can reliably measure nurses’ perception of TCCN in hospital settings when using the ENRS. The research findings can also have an impact on the further development of the ENRS and patterns of knowing in nursing, such as empirical, personal, ethical, and aesthetic, and help us identify and propose strategies to include the language of caring in the ENRS. Given the development and use of the ENRS in healthcare, more research is needed regarding integrating caring and technological competency when using the ENRS, and the PITCCN questionnaire is a tool that can support this research.

### Limitations

This study was conducted with a population of members of nursing teams as participants, where not all of them were educated on the TCCN theory reflected in the items in the PITCCN. Education on TCCN theory would be necessary to improve nurses’ understanding of technological competence and caring integration in the ENRS. In the context of criterion validity, it is worth noting that no comparable questionnaires incorporating caring in nursing, including the use of technology (ENRS), were found. Consequently, a criterion validity assessment was not performed. Exploratory factor analysis and confirmatory factor analysis were not performed due to the limited population size [30] in Slovenian hospitals that use ENRS. ENRS is still being developed in Slovenia, with many hospitals lacking electronic support for nursing care documentation. Consequently, our study concentrated on four hospitals that utilize the same ENRS for nursing care documentation. More than half of the possible population (121 out of 164, 73.78% of surveys were returned) working with this type of information system was collected. We did not observe any direct or indirect impacts of the COVID-19 pandemic on our research. However, it is possible that the pandemic contributed to a lower response rate, as only 121 out of 164 distributed questionnaires were returned.

## Figures and Tables

**Table 1 healthcare-12-00561-t001:** The content validity of the PITCCN_SI.

	Subscale/Item	Number of Experts	Number of Agreements	^a^ ICV-I	^b^ P_c_	^c^ κ*	^d^ Evaluation
	**Training of nurses to provide optimal care**						
1	I continue to consider better care by reflecting on their process of care.	8	5	0.625	0.019	0.618	good
2	I practice like growing up as a nurse.	8	8	1.000	0.000	1.000	excellent
3	I cherish that to convey what I learned from patients and share it with patients.	8	8	1.000	0.000	1.000	excellent
4	I support patients in fulfilling their hopes and desires.	8	8	1.000	0.000	1.000	excellent
5	I communicate their learned experiences of caring for patients with their colleagues and nursing students and share them with them.	8	8	1.000	0.000	1.000	excellent
6	I provide the best nursing care for patients.	8	8	1.000	0.000	1.000	excellent
7	I care for patients considering time and situation.	8	8	1.000	0.000	1.000	excellent
	**Empirical knowledge and whole human knowing**						
8	I use knowledge of the latest clinical pharmacology.	8	8	1.000	0.000	1.000	excellent
9	I use knowledge of anatomy and physiology.	8	8	1.000	0.000	1.000	excellent
10	I use knowledge of well-versed in the state-of-the-art of medical devices in their department. (Empirical knowing)	8	8	1.000	0.000	1.000	excellent
	**Utilisation of information obtained from technology and continuous knowing**						
11	I understand the condition of their patients based on information acquired from technology (ENRS).	8	8	1.000	0.000	1.000	excellent
12	I assess the patient’s condition from information acquired using technology (ENRS).	8	8	1.000	0.000	1.000	excellent
13	I share patient information acquired from technology to illustrate team medical care effectively (ENRS).	8	8	1.000	0.000	1.000	excellent
	**Intentional and ethical nursing of a person**						
14	I encourage patients by caring emotionally.	8	7	0.875	0.005	0.874	excellent
15	I respect patients as unique individuals.	8	8	1.000	0.000	1.000	excellent
16	I know the whole patient.	8	8	1.000	0.000	1.000	excellent
17	I am caring for patients with an unchanged attitude even if they lose their physical functions.	8	8	1.000	0.000	1.000	excellent
18	I behave in ways that can gain the trust of patients.	8	8	1.000	0.000	1.000	excellent
19	I encourage patients by touching their bodies.	8	8	1.000	0.000	1.000	excellent
	**^e^ S-CVI/Ave = 0.974**						

Legend: ^a^ I-CVI (item content validity index) = number giving a rating of 3 or 4/number of experts. ^b^ Pc (probability of a chance occurrence) =
$Pc=(\frac{N!}{A!}\times \left(N-A\right))\times 0.5^{N}$ N = the number of experts, and A = the number agreed on good relevance. ^c^ κ* = kappa designating agreement on relevance: κ* = [I-CVI − Pc]/[1 − Pc]. ^d^ Evaluation criteria for kappa = fair = κ* of 0.40–0.59; good = κ* of 0.60–0.78; and excellent = κ* > 0.78. ^e^ S-CVI/Ave (average scale content validity index) = mean of I-CVI. ENRS = the electronic nursing record system.

**Table 2 healthcare-12-00561-t002:** Descriptive statistics of the sample features (*n* = 121).

Variable		*n*	Valid %	M (SD)
**Sex**	Female	103	85.10	
	Male	18	14.90	
	Total	121		
	Missing values	2		
**Education**	Nursing Assistant	59	48.8	
	Nurse with a diploma degree	53	43.8	
	Nurse with master’s degree	9	7.4	
	Age			36.06 (10.85)
**Years of working** **experiences**	Up to 5 years	49	40.5	
	6–10 years	21	17.4	
	11–20 years	33	27.3	
	21–30 years	5	4.2	
	More than 31 years	13	10.7	
**Use of ENRS**	Up to 5 months	39	32.2	
	6–11 months	47	38.8	
	12–23 months	9	7.4	
	More than 24 months	26	21.5	

Legend: *n* = the number of responses, % = percentage; M = mean, SD = standard deviation. ENRS = the electronic nursing record system.

**Table 3 healthcare-12-00561-t003:** Items of the PITCCN_SI, total variance explained, components, Cronbach’s α, and corrected item-total correlation coefficients for the varimax rotated four-component solution (*n* = 121).

Item	Component				Corrected Item-Total Correlation	Cronbach α If the Item Is Deleted
	1	2	3	4		
1	0.491				0.534	0.919
2	0.699			0.534	0.726	0.914
3	0.786			0.726	0.640	0.916
4	0.824			0.640	0.678	0.915
5	0.782			0.678	0.684	0.915
6	0.749			0.684	0.635	0.916
7	0.700		0.404	0.635	0.676	0.915
8		0.740	0.338	0.676	0.431	0.921
9		0.853			0.505	0.919
10	0.304	0.808		0.431	0.551	0.918
11			0.861	0.505	0.449	0.921
12			0.920	0.551	0.416	0.922
13			0.861		0.491	0.920
14	0.413			0.627	0.739	0.913
15				0.841	0.678	0.915
16				0.847	0.668	0.915
17				0.828	0.615	0.917
18				0.845	0.601	0.917
19	0.314			0.729	0.690	0.915
Total variance explained (%)	43.824	13.030	9.004	7.636		
Cronbach’s α 0.925 (all 19 items)	0.901	0.810	0.897	0.921		

Legend: Components: 1—nursing education to give optimal care; 2—complete human knowing and empirical knowledge; 3—integration of information received from ENRS; 4—committed and ethical nursing care. ENRS = the electronic nursing record system.

## Data Availability

The data supporting this study’s findings are available from the corresponding author upon special request.

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
