# Peer review of "Psychometric Testing of the Slovene Version of the Perceived Inventory of Technological Competency as Caring in Nursing"

_healthcare, 2024, doi:10.3390/healthcare12050561_

Round 1

Reviewer 1 Report (New Reviewer)

Comments and Suggestions for Authors

Dear autors;

Please review the following suggestions

It is a good study. One can see that changes have been made throughout the document to provide adequate theoretical and scientific support for the subject.

The bibliographical review is adequate and provides significant support for the document.

A good validation of the instrument takes into account the experience and training of the panel of experts; one factor to be considered is the equivalence of the semantic structure. The direct validation phase of the culture related to the equivalence of the items should also be taken into account.

As far as the sample is concerned, it is mentioned on page 3, line 141 that 121 people took part in this research. On the one hand, on page 4, line 159, 131 questionnaires were sent out. 10 of these were excluded from the final sample. On the other hand, in the section on the sample (page 6, line 212), 164 questionnaires are mentioned, and it is suggested that this information be checked in order to avoid confusion concerning the data.

There is no mention of a possible direct or indirect impact of the COVID-19 pandemic, given that this instrument was applied to health professionals directly involved in this pandemic.

The discussion is supported by accurate references to the work of authors who support the evidence found. In addition, it is suggested that this section (discussion) does not include statistical data, but rather a comparison of other authors' findings with those found in this paper.

The limitations take into account the limited sample and the underlying reasons.

Yours faithfully

Author Response

Response to Reviewer 1 Comments

1. Summary

We want to thank you for your prompt review of our work. We appreciate the constructive comments and the opportunity to improve our manuscript. In response to your review, we have made great efforts to address all the concerns raised by the reviewer. In the following text, you will find point-by-point responses to the comments from the reviewer. Revisions in the manuscript are shown in red font.

2. Questions for General Evaluation

Reviewer’s Evaluation

Response and Revisions

Does the introduction provide sufficient background and include all relevant references?

Yes/Can be improved/Must be improved/Not applicable

Are all the cited references relevant to the research?

Yes/Can be improved/Must be improved/Not applicable

Is the research design appropriate?

Yes/Can be improved/Must be improved/Not applicable

Our response is in the point-by-point response letter below.

Are the methods adequately described?

Yes/Can be improved/Must be improved/Not applicable

Are the results clearly presented?

Yes/Can be improved/Must be improved/Not applicable

Are the conclusions supported by the results?

Yes/Can be improved/Must be improved/Not applicable

3. Point-by-point response to Comments and Suggestions for Authors

Comments 1: It is a good study. One can see that changes have been made throughout the document to provide adequate theoretical and scientific support for the subject.

The bibliographical review is adequate and provides significant support for the document.

Response 1: We thank the reviewer for the thorough review of our paper, the kind words and the valuable comments that helped us improve our manuscript.

Comments 2: A good validation of the instrument takes into account the experience and training of the panel of experts; one factor to be considered is the equivalence of the semantic structure. The direct validation phase of the culture related to the equivalence of the items should also be taken into account.

Response 2: We thank the reviewer for this comment.

In the pilot study, the research team evaluated the cultural and semantic equivalence and comprehensibility of the items by consulting with experienced nursing team members from the target culture, including registered nurses and nursing assistants. This process aimed to ensure that the translation of the instrument accurately preserved the meaning of each item across the cultures of interest.

Detailed documentation was maintained throughout the translation and adaptation process. It included records of translation, expert reviews, the pilot feedback, and any modifications made to enhance semantic and cultural equivalence. It was reviewed by a panel of experts (PhD-trained professors; one expert in research and methodological studies, two in nursing care, and one in nursing technology). The panel of experts was in contact with the original developers of the PITCCN questionnaire during this part of the process.

The revised manuscript was updated (page 3, red paragraph, and lines 144-148; page 6, and lines 212-216).

Comments 3: As far as the sample is concerned, it is mentioned on page 3, line 141 that 121 people took part in this research. On the one hand, on page 4, line 159, 131 questionnaires were sent out. 10 of these were excluded from the final sample. On the other hand, in the section on the sample (page 6, line 212), 164 questionnaires are mentioned, and it is suggested that this information be checked in order to avoid confusion concerning the data.

Response 3: We thank the reviewer for this comment.

On page 4, line 159, an error occurred that 131 questionnaires were sent. The correct information is that 164 questionnaires were sent.

The revised manuscript was updated (page 4, red paragraph, and line 169).

Comments 4: There is no mention of a possible direct or indirect impact of the COVID-19 pandemic, given that this instrument was applied to health professionals directly involved in this pandemic.

Response 4: We thank the reviewer for this comment.

We did not observe any direct or indirect impacts of the COVID-19 pandemic on our research. However, it is possible that the pandemic contributed to a lower response rate, as only 121 out of 164 distributed questionnaires were returned.

The revised manuscript was updated (page 10, red paragraph, and lines 354-356).

Comments 5: The discussion is supported by accurate references to the work of authors who support the evidence found. In addition, it is suggested that this section (discussion) does not include statistical data, but rather a comparison of other authors' findings with those found in this paper.

Response 5: We thank the reviewer for this comment. We considered your recommendation and deleted the statistical data in the discussion.

Comments 6: The limitations take into account the limited sample and the underlying reasons.

Response 6: We thank the reviewer for this comment.

Reviewer 2 Report (New Reviewer)

Comments and Suggestions for Authors

If you have not conducted Confirmatory factor analysis, because of the unsufficient sample size (for example, less than 10 participants per item in the questionnaire), then I recommend you to quote one or more sources regarding the minimum sample size for CFA as support of what you state in the sentence:

"Exploratory factor analysis and confirmatory factor analysis were not performed due to the limited population size in Slovenian hospitals that use ENRS."

Comments on the Quality of English Language

The sentence in the Abstract is not a full, complete sentence:

"As there were no psychometrically sound instruments to quantify the concept in the Slovene language."

Author Response

Response to Reviewer 2 Comments

1. Summary

We want to thank you for your prompt review of our work. We appreciate the constructive comments and the opportunity to improve our manuscript. In response to your review, we have made great efforts to address all the concerns raised by the reviewer. In the following text, you will find point-by-point responses to the comments from the reviewer. Revisions in the manuscript are shown in red font.

2. Questions for General Evaluation

Reviewer’s Evaluation

Response and Revisions

Does the introduction provide sufficient background and include all relevant references?

Yes/Can be improved/Must be improved/Not applicable

Are all the cited references relevant to the research?

Yes/Can be improved/Must be improved/Not applicable

Is the research design appropriate?

Yes/Can be improved/Must be improved/Not applicable

Are the methods adequately described?

Yes/Can be improved/Must be improved/Not applicable

Are the results clearly presented?

Yes/Can be improved/Must be improved/Not applicable

Are the conclusions supported by the results?

Yes/Can be improved/Must be improved/Not applicable

3. Point-by-point response to Comments and Suggestions for Authors

Comments 1: If you have not conducted Confirmatory factor analysis, because of the unsufficient sample size (for example, less than 10 participants per item in the questionnaire), then I recommend you to quote one or more sources regarding the minimum sample size for CFA as support of what you state in the sentence.

Response 1: We thank the reviewer for this comment.

We added the missing reference.

The revised manuscript was updated (page 10, red paragraph, and line 350).

Comments 2: The sentence in the Abstract is not a full, complete sentence: "As there were no psychometrically sound instruments to quantify the concept in the Slovene language."

Response 2: We thank the reviewer for this comment.

As there were no psychometrically sound instruments to quantify the concept of technological competency as caring in nursing in the Slovene language, we adapted the English version of the questionnaire to the local environment.

The revised manuscript was updated (page 1, red paragraph, and lines 16-17).

Reviewer 3 Report (New Reviewer)

Comments and Suggestions for Authors

Thank you very much for giving me the opportunity to review the manuscript entitled “Psychometric testing of the Slovene version of the Perceived Inventory of Technological Competency as Caring in Nursing”. I have read it with interest. I am not a content expert, and I have checked if the manuscript could be understood by non-expert readers. This study translated the Perceived Inventory of Technological Competency as Caring in Nursing (PITCCN) into Slovene and examined its content and face validity, factor structure, and reliability with the sample of 121 nurses and nurse assistants. The study was well designed and was executed. I believe that the study is informative in that a new, promising instrument to assess technological competency as caring in nursing is developed. I have suggested several minor points authors might want to address.

1) Abstract. It would be helpful if technological competency as caring in nursing practice is defined.

2) Line 65-67, “The questionnaire Perceived Inventory of Technological Competency as Caring and Nursing (PITCCN) was developed by Kato et al. [20], revised by Miyamoto et al. [21], and validated by Ito et al. [22].”. It would be helpful if the factor structure of and the constructs which are measured with this scale are explained.

3) Line 173-174, “Cronbach's alpha was calculated for internal consistency if the coefficient was above 0.7 [28]”. The meaning of this sentence is not clear to me. I would suggest rewriting.

4) Line 179-180, “The reduced variables, called components, are linear combinations of the original variables.”. This sentence is a reputation of the previous sentence. I humbly suggest rechecking.

5) Line 212, “Fully completed 121 out of 164 surveys were returned (return rate = 73.78%).”. I humbly ask authors to check if “164” is correct. Line 159 writes “A total of 131 questionnaires were sent and were to be returned within four weeks”.

6) Line 248-249, “A version of the PITCCN_SI to the original version was obtained, which was semantically and culturally adapted to Slovenian members of nursing teams.”. The meaning of this sentence is not clear to me. I humbly suggest rechecking. 

I hope that my comments will be useful,

Author Response

Response to Reviewer 3 Comments

1. Summary

We want to thank you for your prompt review of our work. We appreciate the constructive comments and the opportunity to improve our manuscript. In response to your review, we have made great efforts to address all the concerns raised by the reviewer. In the following text, you will find point-by-point responses to the comments from the reviewer. Revisions in the manuscript are shown in red font.

2. Questions for General Evaluation

Reviewer’s Evaluation

Response and Revisions

Does the introduction provide sufficient background and include all relevant references?

Yes/Can be improved/Must be improved/Not applicable

Our response is in the point-by-point response letter below

Are all the cited references relevant to the research?

Yes/Can be improved/Must be improved/Not applicable

Is the research design appropriate?

Yes/Can be improved/Must be improved/Not applicable

Are the methods adequately described?

Yes/Can be improved/Must be improved/Not applicable

Are the results clearly presented?

Yes/Can be improved/Must be improved/Not applicable

Are the conclusions supported by the results?

Yes/Can be improved/Must be improved/Not applicable

3. Point-by-point response to Comments and Suggestions for Authors

Comments 1: Abstract. It would be helpful if technological competency as caring in nursing practice is defined.

Response 1: We thank the reviewer for this comment. We have considered your suggestion and added it to the abstract.

It involves integrating the use of technology with the fundamental principles of caring that are central to nursing.

The revised manuscript was updated (page 1, red paragraph, and lines 14-15).

Comments 2: Line 65-67, “The questionnaire Perceived Inventory of Technological Competency as Caring and Nursing (PITCCN) was developed by Kato et al. [20], revised by Miyamoto et al. [21], and validated by Ito et al. [22].”. It would be helpful if the factor structure of and the constructs which are measured with this scale are explained.

Response 2: We thank the reviewer for this comment.

The PITCCN is constituted of four factors, which are measured with this scale: (1) Training of nurses to provide optimal care, (2) Intentional and ethical nursing of person, (3) Utilization of information obtained from technology and continuous knowing, and (4) Empirical knowledge and knowing wholeness of persons.

The revised manuscript was updated (page 2, red paragraph, and lines 69-73).

Comments 3: Line 173-174, “Cronbach's alpha was calculated for internal consistency if the coefficient was above 0.7 [28]”. The meaning of this sentence is not clear to me. I would suggest rewriting.

Response 3: We thank the reviewer for this comment.

Cronbach's alpha was calculated for internal consistency and was used as acceptable if the coefficient was above 0.7 [28].

The revised manuscript was updated (page 4, red paragraph, and lines 183-184).

Comments 4: Line 179-180, “The reduced variables, called components, are linear combinations of the original variables.”. This sentence is a reputation of the previous sentence. I humbly suggest rechecking.

Response 4: We thank the reviewer for this comment. We removed the sentence: "The reduced variables, called components, are linear combinations of the original variables."

The revised manuscript was updated (page 4, red paragraph, and line 188).

Comments 5: Line 212, “Fully completed 121 out of 164 surveys were returned (return rate = 73.78%).”. I humbly ask authors to check if “164” is correct. Line 159 writes “A total of 131 questionnaires were sent and were to be returned within four weeks”.

Response 5: We thank the reviewer for this comment. On page 4, line 164, an error occurred that 131 questionnaires were sent. The correct information is that 164 questionnaires were sent.

The revised manuscript was updated (page 4, red paragraph, and line 169).

Comments 6: Line 248-249, “A version of the PITCCN_SI to the original version was obtained, which was semantically and culturally adapted to Slovenian members of nursing teams.”. The meaning of this sentence is not clear to me. I humbly suggest rechecking.

Response 6: We thank the reviewer for this comment.

A version of the PITCCN_SI was developed, which was semantically and culturally adapted to be used for Slovenian members of nursing teams.

The revised manuscript was updated (page 4, red paragraph, and lines 256-257).

This manuscript is a resubmission of an earlier submission. The following is a list of the peer review reports and author responses from that submission.

Round 1

Reviewer 1 Report

Comments and Suggestions for Authors

1. For whom was the tool evaluated in this study developed? It seems to be a tool developed for nurses, and both nurses and nursing assistants are included in the testing process. The roles of nurses and nursing assistants are different, but I don't understand why they participate as subjects of the same tool. The users of the development tool must be clearly presented.

2. Please describe in more detail in the discussion what the major differences are from the original PITCCN during the verification process.

Author Response

Dear Reviewer.

We want to thank you for your prompt review of our work. We appreciate the constructive comments and the opportunity to improve our manuscript. In response to your review, we have extensively addressed all the concerns. In the following text, you will find point-by-point answers to the comments. Revisions in the manuscript are shown in red font. 

Kind regards.
Yours sincerely.
Cvetka Krel

2. Questions for General Evaluation

Reviewer’s Evaluation

Response and Revisions

Does the introduction provide sufficient background and include all relevant references?

Yes/Can be improved/Must be improved/Not applicable

Are all the cited references relevant to the research?

Yes/Can be improved/Must be improved/Not applicable

Is the research design appropriate?

Yes/Can be improved/Must be improved/Not applicable

Are the methods adequately described?

Yes/Can be improved/Must be improved/Not applicable

Our response is in the point-by-point response letter below.

Are the results clearly presented?

Yes/Can be improved/Must be improved/Not applicable

Are the conclusions supported by the results?

Yes/Can be improved/Must be improved/Not applicable

Our response is in the point-by-point response letter below.

3. Point-by-point response to Comments and Suggestions for Authors

Comments 1: For whom was the tool evaluated in this study developed? It seems to be a tool developed for nurses, and both nurses and nursing assistants are included in the testing process. The roles of nurses and nursing assistants are different, but I don't understand why they participate as subjects of the same tool. The users of the development tool must be clearly presented.

Response 1:

We thank the reviewer for this comment.

We have included nurses and nursing assistants in the study, as in Slovenia, both professional groups are members of nursing teams who actively engage in various nursing interventions and use ENRS within hospital settings. Nursing assistants undergo secondary-level education specific to their role, while nurses receive a 3-year first-cycle Bologna higher education. According to the National Institute of Public Health of the Republic of Slovenia (2020), currently, there are 72% (n = 12.959) nursing assistants and 28% (n = 9.043) nurses in Slovenia.

The revised manuscript was updated (pages 3-4, red paragraph, and lines 144-151).

Comments 2: Please describe in more detail in the discussion what the major differences are from the original PITCCN during the verification process.

Response 2: We thank the reviewer for this comment. Accordingly, we have examined the differences between the original questionnaire PITCCN and PITCCN_SI.

The original instrument had twenty-three items. The questionnaire was tested in Japanese hospitals in intensive care units. Since the original questionnaire also contained questions about managing unconscious patients in intensive care units, we decided to use the newer PITCCN questionnaire by ITO et al. [22], who retained 19 of the 23 items after factor analysis. Thus, we did not include in our research the items of the original PITCCN: 1) I deliberately try to communicate with unconscious patients with the aim of resuscitation 2) I empathize with what patients experience 3) I appreciate knowing that the patient is most hopeful now 4) I am required to respect the privacy of unconscious patients. We found no other differences between the original and PITCCN_SI during the verification process.

The revised manuscript was updated (page 9, red paragraph, and lines 291-300).

4. Response to Comments on the Quality of English Language

Point 1: not assessed

Response 1: /

5. Additional clarifications

/

Reviewer 2 Report

Comments and Suggestions for Authors

This study is relevant because the cultural adaptation of a health measurement tool is always an important contribution for health professionals in the country in question. However, when reading the document, it was possible to see that the wording is not always clear, with not very good quality English, which often makes it difficult to understand what the authors mean.

In the methods, more specifically in the description of the procedures for assessing content validity, it would be good to better understand the training and curriculum of the experts used. 

With regard to the procedures themselves, the description given by the authors seems to refer to what should be done during the development of an assessment instrument - as stated by author 27 - but not in a translation and cultural adaptation situation. This is visible both in the procedures relating to criterion validity but also in internal consistency and criterion validity.

The ethical procedures should also be better explained: the information provided does not seem to be sufficient to guarantee that the anonymity of the participants and the confidentiality of the data has been preserved.

The discussion is poorly explored, as well as being difficult to understand in several sentences due to the quality of the English. Again, in this chapter, there seems to be confusion about the role that the experts called upon to comment on the cultural adaptation of the questionnaire should have: they should comment on the quality of the semantic and cultural equivalence of the items and not on the suitability of the items that make up the original instrument.

From line 234 onwards, it seems that the authors are dedicated to reporting on the instrument and the procedure followed in its development, but they do not discuss the results obtained in this study, so this part of the discussion can be removed as it is not relevant. 

Once again emphasising the importance of this type of study, it should also be noted that its quality is fundamental in order to guarantee that the procedures taken during the cultural adaptation of health measures are rigorous and, as far as possible, uniform.

It is therefore recommended that they be reformulated, taking into account the aspects highlighted.

Comments on the Quality of English Language

Moderate editing of English language required

Author Response

Dear Reviewer.

We want to thank you for your prompt review of our work. We appreciate the constructive comments and the opportunity to improve our manuscript. In response to your review, we have extensively addressed all the concerns. In the following text, you will find point-by-point answers to the comments. Revisions in the manuscript are shown in red font. Please see the attachment.

Kind regards.
Yours sincerely.
Cvetka Krel

2. Questions for General Evaluation

Reviewer’s Evaluation

Response and Revisions

Does the introduction provide sufficient background and include all relevant references?

Yes/Can be improved/Must be improved/Not applicable

Our response is in the point-by-point response letter below.

Are all the cited references relevant to the research?

Yes/Can be improved/Must be improved/Not applicable

Our response is in the point-by-point response letter below.

Is the research design appropriate?

Yes/Can be improved/Must be improved/Not applicable

Our response is in the point-by-point response letter below.

Are the methods adequately described?

Yes/Can be improved/Must be improved/Not applicable

Our response is in the point-by-point response letter below.

Are the results clearly presented?

Yes/Can be improved/Must be improved/Not applicable

Our response is in the point-by-point response letter below.

Are the conclusions supported by the results?

Yes/Can be improved/Must be improved/Not applicable

Our response is in the point-by-point response letter below.

3. Point-by-point response to Comments and Suggestions for Authors

Comments 1: This study is relevant because the cultural adaptation of a health measurement tool is always an important contribution for health professionals in the country in question. However, when reading the document, it was possible to see that the wording is not always clear, with not very good quality English, which often makes it difficult to understand what the authors mean.

Response 1:

We thank the reviewer for the thorough review of our paper, the kind words and the valuable comments that helped us improve our manuscript.

A native speaker of English reviewed the article. We can provide an editing certificate.

Comments 2: In the methods, more specifically in the description of the procedures for assessing content validity, it would be good to better understand the training and curriculum of the experts used.

Response 2: We thank the reviewer for this comment.

Eight experts with a PhD or MSc and over three years of nursing experience assessed content validity. Their educational background encompassed nursing theories, including caring theories, and expertise in utilizing nursing technologies, such as electronic nursing record systems.

The revised manuscript was updated (page 3, red paragraph, and lines 108-111).

Comments 3: With regard to the procedures themselves, the description given by the authors seems to refer to what should be done during the development of an assessment instrument - as stated by author 27 - but not in a translation and cultural adaptation situation. This is visible both in the procedures relating to criterion validity but also in internal consistency and criterion validity.

Response 3: We thank the reviewer for this comment. We have supplemented the methods to ensure the semantic and cultural equivalence of the items.

The semantic equivalence of items was assessed through a combination of forward and back translation processes. The instruments were separately translated into Slovene by two translators. One of the translators was acquainted with the research on technological competency as caring in nursing. Both then discussed the translated instrument with another researcher fluent in English and holds expertise in the research field. A professional translator backtranslated the instrument when preparing a consensus version. The research team and translators compared responses to the translated and original versions of the questionnaire. They eliminated any discrepancies that might indicate problems with semantic or cultural equivalence.

Face validity and the cultural equivalence of items were evaluated in a pilot study involving a convenience sample of 12 experienced members of the nursing teams (Registered Nurses and nursing assistants). They have had more than three years of experience in nursing, have worked in one of the internal wards of Slovenian hospitals, and were proficient in using the ENRS. Participants were invited to complete the questionnaire, estimate the time needed for completion, and assess comprehensibility (face validity) by proposing enhanced phrasing for any unclear items. Following their comments or suggested revisions, the authors engaged in discussions to determine whether the items maintained cultural significance, aiming to reach a consensus before finalizing and confirming the questionnaire.

Detailed documentation was maintained throughout the translation and adaptation process. It included records of translation, expert reviews, the pilot feedback, and any modifications made to enhance semantic and cultural equivalence.

This change can be found in the revised manuscript – page 3, red paragraph, and lines 98-139.

In the context of criterion validity, it is worth noting that no comparable questionnaires incorporating caring in nursing, including the use of technology (ENRS), were found. Consequently, the criterion validity assessment was not performed.

The revised manuscript was updated (page 10, red paragraph, and lines 334-340).

Comments 4: The ethical procedures should also be better explained: the information provided does not seem to be sufficient to guarantee that the anonymity of the participants and the confidentiality of the data has been preserved.

Response 4: We thank the reviewer for this comment.

The questionnaires were distributed in envelopes for the participants to participants who were ready to participate in the research by the head nurse of each ward with the researcher. Data for participants was provided in each questionnaire: “We ask you to participate in the survey, where we assure you that all the information provided will be anonymous and used solely for the study. The results of the data analysis will not reveal your identity or your institution's identity. You agree to participate in the survey by completing and submitting the questionnaire”. The questionnaires had to be completed within four weeks. The participants returned the completed questionnaire in a sealed envelope to the special closed containers given to the researcher by the head nurse. The participant’s completion and return of a survey was treated as implied consent. All questionnaires had security codes attributed to computerised records. The questionnaires are stored in a locked location at the institution where the principal of the researchers is employed for the next ten years, to which only he has access.

The revised manuscript was updated (page 10, red paragraph, and lines 348-364).

Comments 5: The discussion is poorly explored, as well as being difficult to understand in several sentences due to the quality of the English. Again, in this chapter, there seems to be confusion about the role that the experts called upon to comment on the cultural adaptation of the questionnaire should have: they should comment on the quality of the semantic and cultural equivalence of the items and not on the suitability of the items that make up the original instrument.

Response 5: We thank the reviewer for this comment.

We have supplemented the discussion on ensuring the semantic and cultural equivalence of the items.

The research team and translators discussed the translated and original versions of the questionnaire. They eliminated any discrepancies that might indicate problems with semantic or cultural equivalence. To guarantee the questionnaire's suitability in representing the targeted construct, the expert panel, assessing content validity, evaluated the relevance of each question's content, the appropriateness of language, cultural pertinence, and coverage across domains.

The revised manuscript was updated (page 8, red paragraph, and lines 251-256).

Comments 6: From line 234 onwards, it seems that the authors are dedicated to reporting on the instrument and the procedure followed in its development, but they do not discuss the results obtained in this study, so this part of the discussion can be removed as it is not relevant. 

Response 6: We thank the reviewer for this comment.

We deleted the section on reporting on the instrument and its development process.

Comments 7: Once again emphasising the importance of this type of study, it should also be noted that its quality is fundamental in order to guarantee that n. of health measures are rigorous and, as far as possible, uniform.

Response 7: We thank the reviewer for this comment.

A version of the PITCCN_SI to the original version was obtained, which was semantically and culturally adapted to Slovenian members of nursing teams. The process of back translation was used. The forward and backward translations are essential for the survey's validity to achieve semantic equivalence [33] and the high quality of the translation, which confirmed the accuracy of the translations for the Slovene version of the PITCCN questionnaire. The research team and translators discussed the translated and original versions of the questionnaire. They eliminated any discrepancies that might indicate problems with semantic or cultural equivalence. To guarantee the questionnaire's suitability in representing the targeted construct, the expert panel, assessing content validity, evaluated the relevance of each question's content, the appropriateness of language, cultural pertinence, and coverage across domains [26].

We carried out content validity, which is not often used but is recommended for psychometric testing and cultural adaptations [26, 34]. In the professional nursing practice, the differences relate to demographics, culture, and differences in the health care system. Also, the development and use of technology in nursing differ in each country [22, 35]. The CVI indicates that the degree of agreement between the expert raters and the criteria for item acceptability should be no lower than 0.78 [28]. Only one of the items in our study (I continue to consider better care by reflecting on their process of care) in the instrument had an I-CVI score (0.62) lower than 0.78. The experts suggested only a more comprehensible reformulation of the item in the Slovenian language, so it was not eliminated from the questionnaire and was considered in further psychometric testing, where it reached acceptable values when testing internal reliability. The average CVI for the scale was 0.974, showing satisfactory content validity. The content and face validity of PITCCN were found to be adequate. The research team assessed the items' cultural equivalence and comprehensibility by discussing with participants from the target culture (Registered nurses and nursing assistants) in the pilot study to improve clarity and cultural relevance. They confirmed the items were culturally appropriate, easily understood, and free from cultural bias during the pilot study. Comprehensive records were kept during the translation and adaptation process, documenting translation decisions, expert reviews, pilot stady feedback, and modifications made to improve semantic and cultural equivalence. They provide transparency and can facilitate future research or validation efforts [26].

The revised manuscript was updated (page 8, red paragraph, and lines 246-276).

4. Response to Comments on the Quality of English Language

Point 1: Moderate editing of English language required

Response 1: A native English speaker reviewed the article. We can provide an editing certificate.

5. Additional clarifications

/

Reviewer 3 Report

Comments and Suggestions for Authors

Comment to authors

Synopsis: Perceived Inventory of Technological Competency as Caring in Nursing (PITCCN) is an assessment instrument developed for the nursing profession based on a nursing theory that has arisen from technological advancement in modern nursing. This paper reports the validity and reliability data on the Slovene language version of this instrument.

Reviewer's conflict of interest: None

This is a very well prepared manuscript and the contents are thorough. The following revision requests are minor:

  1. Line 78: “… the lack of psychometrically instruments for …” should read “…the lack of psychometrically valid and reliable instruments for …”
  2. 2.1 The translation processes. Remove a period after the “processes” to be consistent with other headings. The back translation by two translators and the process of a consensus version of the translation are well acknowledged.
  3. 2.2 Content validity. Line 103: Please briefly discuss how the level of expertise was documented for the eight experts such as: job title and years of experience on the job. The formula of content validity calculations, criterion values, and references are well written.
  4. Tables: titles, captions, and legends are well organized.
  5. Thank you for including the funding numbers (Line 287) and IRB number (Line 290).
  6. I did not check each reference for accuracy and authenticity.

End of review

Comments on the Quality of English Language

See attached

Author Response

Dear Reviewer.

We want to thank you for your prompt review of our work. We appreciate the constructive comments and the opportunity to improve our manuscript. In response to your review, we have extensively addressed all the concerns. In the following text, you will find point-by-point answers to the comments. Revisions in the manuscript are shown in red font. Please see the attachment.

Kind regards.
Yours sincerely.
Cvetka Krel

2. Questions for General Evaluation

Reviewer’s Evaluation

Response and Revisions

Does the introduction provide sufficient background and include all relevant references?

Yes/Can be improved/Must be improved/Not applicable

Are all the cited references relevant to the research?

Yes/Can be improved/Must be improved/Not applicable

Is the research design appropriate?

Yes/Can be improved/Must be improved/Not applicable

Are the methods adequately described?

Yes/Can be improved/Must be improved/Not applicable

Are the results clearly presented?

Yes/Can be improved/Must be improved/Not applicable

Are the conclusions supported by the results?

Yes/Can be improved/Must be improved/Not applicable

3. Point-by-point response to Comments and Suggestions for Authors

Comments 1: Line 78: “… the lack of psychometrically instruments for …” should read “…the lack of psychometrically valid and reliable instruments for …”

Response 1: We thank the reviewer for this comment. We have considered your suggestion and added it to the manuscript.

The revised manuscript was updated (page 2, red paragraph, and line 78).

Comments 2: 2.1 The translation processes. Remove a period after the “processes” to be consistent with other headings. The back translation by two translators and the process of a consensus version of the translation are well acknowledged.

Response 2: We thank the reviewer for this comment.

We have considered your suggestion.

The revised manuscript was updated (page 2, red paragraph, and line 93).

Comments 3: 2.2 Content validity. Line 103: Please briefly discuss how the level of expertise was documented for the eight experts such as: job title and years of experience on the job. The formula of content validity calculations, criterion values, and references are well written.

Response 3: We thank the reviewer for this comment.

Eight experts with a PhD or MSc and over three years of nursing experience assessed content validity. Their educational background encompassed nursing theories, including caring theories, and expertise in utilizing nursing technologies, such as electronic nursing record systems.

The revised manuscript was updated (page 3, red paragraph, and lines 108-111).

Comments 4: Tables: titles, captions, and legends are well organized.

Response 4: We thank the reviewer for this comment.

Comments 5: Thank you for including the funding numbers (Line 287) and IRB number (Line 290).

Response 5: We thank the reviewer for the thorough review of our paper and the kind words.

Comments 6: I did not check each reference for accuracy and authenticity.

Response 6: We thank the reviewer for this comment.

We carefully checked each reference for accuracy and authenticity.

4. Response to Comments on the Quality of English Language

Point 1: Minor editing of English language required

Response 1: A native English speaker reviewed the article. We can provide an editing certificate.

5. Additional clarifications

/
